



# A key factor initiating surface ablation of Arctic sea ice: Earlier and increasing liquid precipitation

Tingfeng Dou,[1,2] Cunde Xiao,[3,2,4]* Jiping Liu,[5] Wei Han,[6] Zhiheng Du,[2] Andrew R. Mahoney,[7] Joshua Jones,[8] Hajo Eicken[8]

[1]College of Resources and Environment, University of Chinese Academy of Sciences, Beijing 100049, China.

[2]State Key Laboratory of Cryospheric Sciences, Cold and Arid Regions Environmental and Engineering Research Institute, Chinese Academy of Sciences, Lanzhou 730000, China.

[3]State Key Laboratory of Earth Surface Processes and Resource Ecology, Beijing Normal University, Beijing 100875, China.

[4]Institute of Polar Meteorology, Chinese Academy of Meteorological Sciences, Beijing 100081, China.

[5]Department of Atmospheric and Environmental Sciences, University at Albany, State University of New York, Albany, NY, USA.

[6]Beijing Meteorological Observation Center, Beijing 102600, China.

[7]Geophysical Institute, University of Alaska Fairbanks, Fairbanks, AK 99775-7320, USA.

[8]International Arctic Research Center, University of Alaska Fairbanks, Fairbanks, AK 99775-7340, USA.

*Correspondence to*: Cunde Xiao (cdxiao@bnu.edu.cn)

**Abstract. Snow plays an important role in the Arctic climate system, modulating heat transfer in terrestrial and marine environments and controlling feedbacks. Changes in snow depth over Arctic sea ice, particularly in spring, have a strong impact on the surface energy budget, influencing ocean heat loss, ice growth and surface ponding. Snow**
**conditions are sensitive to the phase (solid or liquid) of deposited precipitation. However, variability and potential trends of rain-on-snow events over Arctic sea ice and their role in sea-ice losses are poorly understood. Time series of surface observations at Utqiaġvik, Alaska reveal rapid reduction in snow depth linked to late-spring rain-on-snow events. Liquid precipitation is key in preconditioning and triggering snow ablation through reduction in surface albedo as well as latent heat release determined by rainfall amount, supported by field observations beginning in 2000**
**and model results. Rainfall was found to accelerate warming and ripening of the snow pack, with even small amounts (such as 0.3 mm recorded on May 24th, 2017) triggering the transition from the warming phase into the ripening phase. Subsequently, direct heat input drives snow melt, with water content of the snow pack increasing until meltwater output occurs, with an associated rapid decrease in snow depth. Rainfall during the ripening phase can further raise water content in the snow layer, prompting onset of the meltwater output phase in the snow pack. First**
**spring rainfall in Utqiaġvik has been observed to shift to earlier dates since the 1970s, in particular after the mid-**





**1990s. Early melt season rainfall and its fraction of total annual precipitation also exhibit an increasing trend. These changes of precipitation over sea ice may have profound impacts on ice melt through feedbacks involving earlier onset of surface melt.**

## 1    Introduction

Arctic sea ice has been experiencing rapid decline in both extent and thickness in recent decades (Stroeve et al., 2007, 2012; Comiso et al., 2008). The ten lowest sea ice extent anomalies in record have all occurred in recent decade. Sea-ice thinning trends (Kwok et al., 2009, 2011) have been associated with first-year sea ice replacing thicker multi-year sea ice (Maslanik et al., 2007, 2011; Giles et al., 2008). These changes make Arctic sea ice more susceptible to variations in thermodynamic forcings, increasing interannual variability (Kwok et al., 2009; Maslanik et al., 2007, 2011; Nghiem et al., 2007; Notz et al., 2009; Laxon et al., 2013). Snow over sea ice plays an important role in the growth and melt of Arctic sea ice (Maykut et al., 1971; Maykut, 1986; Blazey et al., 2013; Perovich et al., 2012). Snow has a lower thermal conductivity and higher albedo (~0.7-0.9 for wet to dry snow) than sea ice, which limits the absorption of solar energy by sea ice as well as by the ocean beneath sea ice (Eicken et al., 2004; Perovich et al., 2012). Screen and Simmonds (2012) showed that the fraction of Arctic summer precipitation occurring as snow has declined in recent decades due to lower-atmosphere warming, and this change of precipitation has likely contributed to the decrease in sea ice extent by reducing the area of snow-covered ice and the resulting surface albedo during summer.

In spring, changes in the amount of snow can either curb or foster sea ice melt. Thick snow helps maintain high surface albedos during the melt season (Eicken et al., 2004), and reduces solar heating of ice and upper ocean (Sturm et al., 2002). In contrast, thin snow melts back earlier in spring and promotes the formation of melt ponds (Eicken et al., 2004; Petrich et al., 2012), which absorb approximately 1.7 times more solar radiation than bare ice and approximately 5 times more than cold, snow-covered sea ice (Perovich et al., 2002, 2012; Webster et al., 2014), accelerating ice decay and solar heating of the upper ocean in spring.

Spring snow depth on sea ice is very sensitive to the phase of precipitation. Solid precipitation increases snow depth, protecting sea ice from melt. Conversely, liquid precipitation heats the snow pack, changes snow grain morphology and lowers albedo, decreasing snow depth. Data from Operation IceBridge flights (2009-2013) indicate an average snow depth on Arctic sea ice of ~20 cm (22.2 cm in the western Arctic and 14.5 cm in the Beaufort and Chukchi Seas) (Webster et al., 2014), which renders the thin sea-ice snowpack particularly susceptible to earlier surface ablation and shorter duration as a result of liquid precipitation. An assessment based on 37 state-of-the-art climate models indicated that in the future rain is projected to become the dominant form of precipitation over the Arctic region (Bintanja and Andry, 2017). Rain-on-snow events over Arctic sea ice are likely to have profound impacts, particularly in late spring when the snowpack has warmed. However, to date no such investigation has been completed over Arctic sea ice.



Here, we investigate the role of liquid precipitation in initiating snow melt and the sea ice ablation season over sea ice based on field measurements in the coastal Chukchi Sea. An energy balance model was adopted to help develop a mechanistic interpretation of the observations. The variability of rain-on-snow events over sea ice and the timing of first spring rain are

5 analyzed using long-term meteorological records available at Utqiaġvik, Alaska.

## 2 Data and Methodology

### 2.1 Data

**Air Temperature and Snow Depth at MB Site**

A snow and ice mass Balance (MB) site was deployed on undeformed landfast first-year ice in the Chukchi Sea north of Utqiaġvik. At this location the ice is homogeneous, and it forms primarily through in-situ freezing rather than advection and deformation and provides ice and snow data representative of level, undeformed ice (Druckenmiller et al., 2009). Snow depth was measured with a Campbell SR50 sonic ranger fixed to a mast extending through the ice. The accuracy is about ±1 cm. Air temperature was measured 2 m above the ice with a shielded Campbell CS500 sensor in 2013, and with a

Campbell HMP155A from 2014 to 2016. In 2017, a shielded RoTronic HC2S3 was deployed 2.2 m above the initial snow surface. Data were recorded every 15 minutes and transferred via ftp to University of Alaska Fairbanks where they were processed (Druckenmiller et al., 2009; Eicken et al., 2012). We used the data for 2013-2017 (https://arcticdata.io/catalog/#view/doi:10.18739/A2D08X).

**Wind and Relative Humidity at MB Site**

The relative humidity was measured with a Campbell CS500 instrument in 2013, and a Campbell HMP155A instrument from 2014 to 2016. The shielded RoTronic HC2S3 that measured air temperature in 2017 also measured relative humidity. Wind direction and speed was measured by two RM-Young/Campbell 5108-L anemometers, one 2.1 m above the initial snow surface and the other 4.1 m above the initial snow surface.

**Radiation, Albedo and Surface Temperature near MB Site**

From April through June 2017, we conducted radiation and surface albedo measurements at a site near MB. The distribution of snow depth at this location is the same as that at MB. Radiation was measured using a CNR4 net radiometer that records the upwelling and downwelling shortwave and longwave radiation. The surface albedo was derived from the upward solar radiation dividing by the incident solar radiation. The surface temperature was measured with a SI-111 infrared radiometer. The sensors were fixed on a bracket 1.5m above the ground. Data were recorded every 5 minutes and collected by the

LoggerNet 4.0 (CR1000).

**Air Temperature and Precipitation at Utqiaġvik WSO AP Station**

The data analyzed here comprises daily precipitation and snowfall from January 1952 to June 2017 for the Utqiaġvik Weather Service Office airport weather station (WSO AP), located near the coast of the Chukchi Sea at Utqiaġvik (available from the Alaska Climate Research Center, http://climate.gi.alaska.edu/acis_data). The snowfall data is given as snow water





equivalent (cm-we). The snowfall amount is subtracted from the total precipitation to obtain the rainfall amount (also in units of cm-we).

## 2.2 Methodology

### Modelling of Snow Depth, Snow Density and SWE

The surface energy balance for the snow pack overlying sea ice can be defined as:

$$Q_{net} = Q^* + Q_s + Q_l + C + R \tag{1}$$

where $Q_{net}$ is the net energy flux at the snow pack on sea ice, $Q^*$ is the net radiative flux, $Q_s$ is the turbulent sensible heat flux, $Q_l$ is the turbulent latent heat flux, $C$ is the conductive heat flux, and $R$ is the heat input by rain. The net radiative flux is composed of the net shortwave and longwave components which are derived from the observed incoming and outgoing

radiative fluxes with a CNR4 net radiometer (see more details in the data section). The sensible heat $Q_s$ and latent heat $Q_l$ were calculated by:

$$Q_s = -\rho_a c_p C_H (T_s - T_a) V_z \tag{2}$$

$$Q_l = -\rho_a r_l C_E (q_s - q_a) V_z \tag{3}$$

where $V_z$ is the mean wind speed in one hour at a height z, and $\rho_a$ and $c_p$ denote the air density and the specific heat capacity

of air. $r_l$ is the vaporization enthalpy. $(T_s - T_a)$ and $(q_s - q_a)$ are the differences in temperature and specific humidity between the snow surface and atmosphere at the height z, respectively. $C_H$ and $C_E$ are the bulk transfer coefficients estimated from a simple non-iterative algorithm (Launiainen et al., 1995) based on the Monin-Obukhov similarity theory.

The conductive heat flux $C$ was estimated as:

$$C = -k(T_s - T_i)/H_s \tag{4}$$

$$k = 0.138 - 1.01\rho_s + 3.233\rho_s{}^2 \quad \{0.156 \le \rho_s \le 0.6\} \tag{5}$$

$$k = 0.023 - 1.01\rho_s + 0.234\rho_s{}^2 \quad \{\rho_s \le 0.156\} \tag{6}$$

where $k$ is the thermal conductivity of snow, and $T_s$ is the snow surface temperature. The observed ice surface temperature $T_i$ was applied in this study. $H_s$ is snow depth. $k$ varies with snow density $\rho_s$ according to the regression Eq. (5) and Eq. (6) as

suggested by Sturm et al. (2002).

For heat input by rain, two settings need to be considered. If rain falls on snowpack that is at the freezing point,

$$R = \rho_w C_w r(T_r - T_m) \tag{7}$$

where $C_w$ is the heat capacity of water, r is the rainfall rate (m s$^{-1}$) and $T_r$ is the rain temperature. Rain is cooled to the

freezing point, giving up sensible heat to warm the snowpack .

If rain falls on a snowpack below the freezing point,

$$R = \rho_w C_w r(T_r - T_m) + \rho_w L_m r \tag{8}$$





Where $L_m$ is the latent heat of fusion, and $T_m$ is the freezing point. Rain first cools to the freezing point, giving up sensible heat. Thereafter the rain will freeze, releasing latent heat, which can heat the snowpack very effectively.

Once the snowpack reaches warming phase, the positive energy budget is used to melt snow. The amount of snow melt ΔSWE (snow water equivalent in m) was estimated as:

$$\Delta\text{SWE} = -Q_{net}/(\rho_w L_m) \tag{9}$$

where $Q_{net}$ is the net energy flux derived from Eq. (1), $\rho_w$ denotes water density in units of kg m$^{-3}$.

The snow density changes were modelled based on:

$$\Delta\rho_s = \rho_s C_1 SWE\, exp(-C_2\rho_s)exp(-0.08(T_0 - T_a))\Delta t \tag{10}$$

Where $C_1$ and $C_2$ are empirical coefficients, which are 7.0 m$^{-1}$ h$^{-1}$ and 21.0 cm$^3$ g$^{-1}$ according to the field measurements in

Yen et al. (1981). $T_0$ is the freezing point temperature. Here, $\Delta t$ equals to one hour.

The snow depth $H_s$ was modeled by:

$$H_s = \frac{(SWE+\Delta SWE)\rho_w}{\rho_s} + SWE_{new}\rho_w/\rho_{snew} \tag{11}$$

Where $SWE$ is snow water equivalent of snow cover in the units of m, $\rho_s$ is snow density in units of kg m$^{-3}$, $SWE_{new}$ is new

deposited snow in snow water equivalent. The density of new fallen snow $\rho_{snew}$ is 102 kg m$^{-3}$ in average derived from the field measurement in Chukchi Sea 2017.

The energy required to reach the isothermal state is calculated according to [$c_i.\rho_w.SWE.(T_{ave}-T_m)$] by Dingman et al. (2015), where $C_I$ is the heat capacity of snow/ice (2.1 kJ kg$^{-1}$ °C$^{-1}$), $T_{ave}$ is the average temperature of the snowpack, $T_m$ is the freezing

point of snow, $\rho_w$ is the density of water, and $SWE$ is the snowpack water equivalent in meters. The temperature profile of the snowpack used to track the timing of the isothermal state was measured with a CRREL-designed thermistor string employing Beaded Stream thermistors. The thermistors were spaced 2 cm apart, and measure temperature with 0.1°C accuracy.

**Model Experiments**

Two experiments were conducted to quantitatively estimate the contribution of rain to snow ablation. In the control run, we applied the meteorological observations to drive the model and simulate the snow depth, snow density and SWE. These observations include wind, air temperature, relative humidity, snow surface temperature, upward and downward longwave radiation, incoming solar radiation, surface albedo, rainfall, snowfall, snow temperature and snow-ice interface temperature.

The measured snow depth and snow density were used to validate the model results. In the sensitivity experiment, we



excluded the impacts of rain by lowering albedo and eliminating the latent heat and sensible heat terms contributed by rainfall.

As observed in this study and previous studies, such as Perovich et al. (2002, 2012, 2017), rain can decrease the surface albedo by ~0.1 within a few hours. This impact on albedo is quite different from that of a gradual warming/melting process. The latter needs ~10 days to reduce the albedo by the same amount (Perovich et al., 2002, 2017). In the sensitivity experiment, we derived an evolutionary sequence of albedo without rain based on a simplifying assumption, in which albedos are linearly extrapolated based on the observations of previous three days using the method of Perovich et al. (2017) for the period May 24$^{th}$ - June 3$^{rd}$ (also see Fig. 2 and Fig. 3 in the reference). In contrast, the control experiment includes all impacts of rainfall on surface properties and fluxes, and therefore drawing on the observed albedo time series from the same period. The observed downwelling solar irradiance was applied to calculate the absorbed solar radiation with and without rain.

### Significance Testing

We calculated the significance value of a linear trend for first rainfall date, rainfall in May, total precipitation and rainfall-total-precipitation ratio in May using a Student's t-test. The trend is significant when $p \leq 0.05$ with 95% confidence.

### 3    Observed Evidence of Rapid Reduction in Snow Depth Associated with Liquid Precipitation

Field measurements at a mass balance site (MB site) on landfast sea ice near Utqiaġvik, Alaska in April-June from 2013 to 2015, revealed rapid declines in snow depth once non-freezing rain fell on the snow. Figure 1 shows the variations of snow depth and surface air temperature observed in 2013-2015. It appears that snow depth on sea ice started to decrease when air temperature rose above the freezing point (0°C). Snow depth then decreased sharply and persistently during subsequent days (6, 3, and 6 days for 2013, 2014, and 2015, respectively). The change in surface air temperature itself cannot explain such rapid reduction in snow depth since surface air temperature fluctuates above and below the freezing point at this time. Rather, the first non-freezing rain events of the year that was immediately followed by the rapid decrease in snow depth might be responsible for such phenomenon. Our available observations from prior years at Utqiaġvik and in the ice pack of the Chukchi Sea corroborate these findings (Fig. 2), as do studies suggesting that the transition into the Arctic surface melt season is linked to pronounced synoptic events, rather than through gradual heating processes (Alt et al., 1987; Persson et al., 1997; Stone et al., 2002; Wang et al., 2005; Sharp et al., 2009; Persson et al., 2012).

### 4    Observations and Model Simulations of Key Processes

A primary mechanism for acceleration of surface melt and ablation is the rain-induced rapid lowering of surface albedo. In order to evaluate this impact, we conducted field measurements of surface albedo in conjunction with characterization of the state of the snow and ice cover on Chukchi Sea ice in April and June, 2017. Observations showed that surface albedo decreased sharply on May 24$^{th}$, 25$^{th}$ and 27$^{th}$ by 0.12, 0.10 and 0.13, respectively, coinciding with the occurrence of rain-on-





snow events (Fig. 3). The observed snow morphology (Fig. 4) and water content (Table 1) indicated significant melt of the snowpack after rainfall for these three days, corresponding to a decrease in snow depth (Fig. 5). Comparison with the earlier two warming events (May 15[th] and 18[th]), with temperatures at or above the freezing point, demonstrates that the warming events alone did not result in such a rapid decrease in albedo, but that liquid precipitation plays a key role. Consistently, two

5    earlier studies also supported that a sharp drop in surface albedo by over 0.05 within a single day was associated with a rain on snow event, different from the gradual decline in surface albedo associated with seasonal surface warming and individual warming events (Perovich et al., 2002, 2017). Such rapid decrease in surface albedo may result in a significant increase in the absorbed shortwave flux. In addition, the rain can directly bring heat into the snow layer, and heat the snow pack interior through the release of latent heat during refreezing of rainwater in the early stages of snow warming.

In order to quantitatively estimate the contribution of liquid precipitation towards rapid decreases in snow depth, we consider three basic snowmelt phases: warming, ripening, and meltwater output phase (Dingman, 2015). For the warming phase, the absorbed energy raises the average snowpack temperature to the freezing point and the snowpack becomes isothermal. Only in an isothermal snowpack is the absorbed energy transformed effectively into snow melt, initiating the snowpack ripening

phase, which in turn leads into the meltwater output phase.

Based on our latest and most comprehensive field measurements, the surface energy budget and contribution from each component were estimated, to identify the dominant factors governing the warming phase of snow melt during three key periods with rainfall occurrence. The first rainfall in 2017 was recorded as starting at 10:00am on May 24[th] (the average

snow cover temperature was -0.7°C). The observed snow temperature showed that the upper layers of the snow pack (16cm) became isothermal after two hours. The observed snow particle size and water content data indicate that the upper 10cm began to melt immediately (Fig. 4 and Table 1). Interestingly, the modeled short-wave radiation absorbed by the snow layer without rainfall (405 KJ m[-2]) could offset the heat loss (-391 KJ m[-2]) from long-wave radiative loss and heat conduction, but the residual – which includes latent and sensible heat - was too small to increase the temperature of the snow layer to the

freezing point. During this period, rain changed the energy balance, initiating the warming phase of snow melt in two ways: 1) increasing the absorption of solar radiation (275.47 KJ m[-2]) by lowering surface albedo; 2) transferring heat into the snow pack. At the same time, such rain events may exceed the storage capacity of water in the snow pack since the snow temperature was still low at this time. As a result, water drains downward, forming ice layers in the lower part of the snow pack and releasing latent heat (contributing in total 50.5 KJ m[-2]). Therefore, rainfall is believed to be the main factor in

rapidly warming the snow layer to an isothermal state in this case.

From the night of May 24[th] to the morning of May 25[th], the snow temperature fell below the freezing point (-1.0°C), and then rainfall occurred at 4:00 am on May 25[th]. The snow temperature observations demonstrated that the snow layer reached an isothermal state after five hours. During this period, the solar radiation absorbed by the snow cover would have been 128 KJ




m$^{-2}$ if there were no rainfall, which is not enough to make up for the total energy loss (171 KJ m$^{-2}$) mainly from the long-wave loss (-83 KJ m$^{-2}$), and from sensible (-36 KJ m$^{-2}$) and latent heat transfer (-37 KJ m$^{-2}$) and heat conduction (-14 KJ m$^{-2}$). Due to the occurrence of rainfall and the resulting albedo reduction, the snow pack absorbed an additional 97 KJ m$^{-2}$ of solar radiation, and the rain also brought 198 KJ m$^{-2}$ into the snow pack, mainly through latent heat release and direct heat input.

Most of the rain-induced energy transfer was used to warm the snowpack, once the snow pack reached the warming phase, the remaining energy was used to melt the snow further, pushing the snowpack into ripening phase.

A heavy snowfall occurred during the evening of May 25$^{th}$ through the morning of May 26$^{th}$. A constant SWE and reduced snow thickness indicated that there was significant snow densification during the daytime on May 26$^{th}$, which was confirmed

by the increase in snow density (Fig. 5). From the evening of May 26$^{th}$ to the morning of 27$^{th}$, the snow temperature decreased to -2.9°C. Subsequently, rainfall occurred at 5:00 am on May 27$^{th}$, and the entire snow pack reached an isothermal state within 5 hours after the rainfall, as observed in the snow temperature record. During this period, the heat loss from long-wave radiation was larger than other components of the heat budget (-795 KJ m$^{-2}$). The absorbed solar radiation (479 KJ m$^{-2}$), latent heat (31 KJ m$^{-2}$), and sensible heat (91 KJ m$^{-2}$) were far from enough to offset this part of the energy budget

in the absence of rainfall. The rainfall contributed 390 KJ m$^{-2}$ to the energy balance by reducing the surface albedo, and contributed 318 KJ m$^{-2}$ by bringing heat directly into the snow pack and releasing latent heat (the latter accounted for the main contribution).

The model results shown above demonstrate that liquid precipitation can lead to completion of the warming phase within

several hours, subsequently initiating the melt season (Fig. 5 and Table 2). Once the warming phase is reached, the remaining energy is used to further melt the snow, producing significant meltwater flow and contributing to snowpack ripening, together with the subsequent absorption of solar radiation (some of which also contributed from rain-on-snow). According to Table 2, the remaining energy was 377 KJ m$^{-2}$ on May 24$^{th}$. For this period, 534 KJ m$^{-2}$ was needed to push the snow pack into the ripening phase. The remaining energy contributed substantially to attainment of the ripening phase,

which lasted only briefly on May 24$^{th}$ due to rapid warming; on the 25$^{th}$, the remaining energy was 143 KJ m$^{-2}$, and on the 27$^{th}$, it was 86 KJ m$^{-2}$. Subsequently, the absorbed energy drove further snow melt, with water content of the snow pack increasing until meltwater output occured, with an associated rapid decrease in snow depth. If rainfall occurs during the ripening phase, it increases water content in the snow layer, pushing the snow pack into the meltwater output phase. This is confirmed by the model simulations showing that SWE decreased significantly within a few hours after each rainfall. In the

absence of rainfall, warming is mostly sluggish, and the snow depth reduction is much more gradual as snow melt proceeds, as was the case in 2002 (Fig. 2). A comparison of modeled SWE for the cases with and without rainfall demonstrates that in the case where the long-wave radiant flux is kept consistent with the observations, in the absence of rain, the snow pack does not undergo such rapid ablation (Fig. 5).



## 5 Variability of Rain-on-snow Events

Having demonstrated the profound impact of rainfall on snow depth and ablation, we explore variability in the timing of first rain-on-snow events since the start of the available record. Due to the lack of long-term continuous observations over sea ice, we employ observations from Utqiaġvik WSO AP station, which is close to the MB site. Precipitation and surface air temperature have been measured at WSO AP since 1902 with large data gaps prior to 1952. Here we use air temperature and precipitation data from 1952 to 2017. A comparison of surface air temperature between WSO AP and MB site shows close correspondence (Fig. 6). The amount of liquid precipitation was not recorded at the MB site, but we did record the timing of rainfall on sea ice or at the laboratory near the MB site from April through June in the field expedition of 2015 and 2017. This timing is in good agreement with the records from WSO AP. Hence, meteorological conditions at WSO AP are representative of the MB site.

As shown in Figure 7, the first rain-on-snow events of the year have been shifted to earlier dates over the past 60 years (2.8 days per decade, P<0.01). This trend towards earlier spring rainfall is more pronounced since the early 2000s (26.9 days per decade during 2000-2015, P<0.01), which is consistent with the accelerated decline of Arctic sea ice since the early 2000s. Meanwhile, the timing of surface air temperature rising above the freezing point also occurs earlier for the past 60 years (3.0 days per decade, P<0.01, Fig. 7). There is a clear relationship between the timing of first rainfall and the timing of air temperature rising above the freezing point (r=0.66). After removing the linear trend, the correlation is 0.57 (p<0.01). On average, the timing of air temperature rising above the freezing point is earlier than the first rainfall event by 9.1 days, suggesting that air temperature exceeding the freezing point is not in of itself a driver of rain-on-snow events. Further analysis indicates that in some years (32%), after the warming events continued for 1-2 days, air temperature dropped again without occurrence of recorded rainfall. Similarly, warming events of 3 days duration without rainfall account for 21 % of all cases. Hence, we re-calculate the timing of warming events that persisted for at least 4 days. Results show that these two measures of spring warming are highly positively correlated (r=0.96). After removing the linear trend, the correlation is still strong (r=0.95, p<0.01), suggesting that the year-to-year variability of the timing of first spring rainfall is closely tied to the timing of persistent warming events (Fig. 7), which might be associated with large-scale weather events.

Prior to the mid-1990s there was almost no rainfall in May (Fig. 8). Since then, the amount of rainfall has increased, especially in the past 10 years. Rainfall amount in May has been increasing significantly over the past 60 years (Fig. 9), with a linear trend of 0.43 mm per decade during 1952-2015 (p<0.01) and 1.4 mm per decade since the mid-1990s (p<0.01). By contrast, the total precipitation has not changed significantly before and after the mid-1990s, but has increased substantially over the past few years (Fig. 8). The trend towards higher ratios of rain-to-total precipitation (R-P ratio) in May is significant over the past 60 years (0.04 per decade, p<0.01, Fig. 8), especially after the mid-1990s (0.09 per decade, p<0.01).

## 6 Discussion and Conclusions





Snow on sea ice strongly impacts the surface energy budget, driving ocean heat loss, ice growth and surface ponding. While the role of snow depth and snowfall variations is well understood, this study demonstrated that rain on snow events are a critical factor in initiating the onset of surface melt over Arctic sea ice, primarily through reduction in surface albedo as well as latent heat release. By pushing the snow pack into the isothermal, ripening and meltwater output phase, liquid

precipitation can sharply reduce snow depth and initiate the onset of rapid surface ablation. The increases in downwelling longwave fluxes through cloud warming associated with rainfall events contribute to warming of the snow pack to the melting point, but are not sufficient to drive the temperature of the entire snow layer into an isothermal state on short time scales. In contrast, the occurrence of liquid precipitation can induce a quick transition of the snow temperature from diurnally varying to an isothermal state. The observations at Utqiaġvik and in the offshore Chukchi Sea ice pack suggest that

at least in some years rain on snow events act as an effective, mostly irreversible trigger for the transition into the surface ablation season. In cases where melt onset occurred in the absence of rain, increases in downward longwave fluxes largely offset the longwave radiation heat loss of snow and are thus key to melt initiation (Mortin et al., 2016). However, as shown from our observations and model results, snow melt triggered by increases in net long-wave radiation is much slower than that driven by liquid precipitation.

This study for the first time assembles process studies and long-term observations at an important coastal site in North America, showing that onset of spring rainfall over sea ice has shifted to earlier dates since the 1970s, in particular since the mid-1990s. Early melt season rainfall and its fraction of total annual precipitation also exhibit an increasing trend. Based on the observational evidence and model results, we speculate that earlier and increasing liquid precipitation leads to earlier and

20 more rapid melt of snowpack over sea ice, allowing for earlier formation of melt ponds. This strengthens the ice-albedo feedback, leading to greater ice mass loss in summer (Perovich et al., 1997; Stroeve et al., 2014) with the resulting thinner ice in turn reducing the ice pack in September (Notz, 2009). This study deepens the understanding of the trigger mechanism of sea ice ablation, which is helpful in improving the modeling and seasonal prediction of Arctic sea ice extent.

**Author contribution:** T. Dou, C. Xiao, J. Liu and H. Eicken jointly conceived the study and wrote the manuscript with additional input from W. Han, Z. Du, A. Mahoney and J. Jones. A. Mahoney, H. Eicken and T. Dou conceived field measurements, generated in-situ data and associated products used in this study. T. Dou performed the analyses. All of the authors discussed the results and contributed to interpretations.

*Acknowledgments.* This study is funded by the National Key Research and Development Program of China (2018YFC1406100), the National Nature Science Foundation of China (NSFC, 41425003, 41401079), the key project of CAMS: Research on the key processes of Cryospheric rapid changes (KJZD-EW-G03-04) and the Climate Program Office, NOAA, U.S. Department of Commerce (NA15OAR4310163 and NA14OAR4310216). MB site data at Utqiaġvik were collected through support of the U.S. National Science Foundation, grant PLR- 0856867. We thank Chris Polashenski and



Nicholas Wright providing the snow temperature profile data at MB in 2017, funded under NSF Grant ARC-1603361, The Snow, Wind, and Time Project.

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





**Table 1: Water content of snowpack in different depth observed over Chukchi sea north Utqiaġvik during May 23[rd] through May 27[th], 2017.**

| Date (local time) / Snow depth (cm) | May 23[rd] | May 24[th] | | May 25[th] | | May 26[th] | May 27[th] |
|---|---|---|---|---|---|---|---|
| | 14:45 | 14:50 | 16:55 | 15:30 | 17:10 | 15:30 | 15:00 |
| 2.5 | 1.7 | 5.8 | 5.3 | 4.8 | 6.0 | 2.3 | 12.3 |
| | 1.8 | 5.8 | 5.5 | 5.0 | 6.0 | 2.4 | 12.3 |
| | | 5.5 | 6.4 | 4.7 | 5.1 | 1.9 | 12.3 |
| | | 5.0 | | 4.4 | | 2.4 | 10.4 |
| 5.0 | 3.2 | 3.5 | 2.7 | 5.8 | | 1.8 | 11.1 |
| | 2.4 | 3.5 | 2.8 | 5.6 | | 1.4 | 10.5 |
| | 3.6 | 3.4 | 3.4 | 5.8 | | 1.7 | 5.5 |
| | | 3.7 | | 5.7 | | 2.0 | 6.7 |
| 7.5 | 2.1 | | 2.7 | 5.0 | 4.7 | 3.8 | 11.3 |
| | 1.7 | | | 5.4 | 5.1 | 4.0 | 10.9 |
| | 2.1 | | | 5.4 | | 4.4 | |
| | | | | 5.7 | | 5.6 | |
| 10.0 | | | | 5.9 | | 4.4 | 8.7 |
| | | | | 2.3 | | 5.3 | 7.6 |
| | | | | | | 4.7 | |
| | | | | | | 4.6 | |




**Table 2: Components of the surface energy budget, required for snow pack to reach the isothermal state (warming phase).**

| Date | Modelled latent heat (KJ m$^{-2}$) | Modelled sensible heat (KJ m$^{-2}$) | Modelled heat conductive flux (KJ m$^{-2}$) | Observe net longwave radiation (KJ m$^{-2}$) | Absorbed solar radiation without rain (KJ m$^{-2}$) | Energy required to reach the isothermal state (KJ m$^{-2}$) | Heat brought into the snow by rain(KJ m$^{-2}$) (Direct heat input + latent heat) | Absorbed solar radiation due to reduced albedo by rain(KJ m$^{-2}$) | Observed net solar radiation(KJ m$^{-2}$) |
|---|---|---|---|---|---|---|---|---|---|
| 5/24/2017 10:00 | 0.3 | 2.0 | -3.7 | -34.6 | 98.3 | | 11.5 | 22.6 | 120.8 |
| 5/24/2017 11:00 | 5.3 | 7.0 | -4.3 | -121.8 | 100.3 | | 39.1 | 85.7 | 186.0 |
| 5/24/2017 12:00 | -1.4 | -1.3 | -4.4 | -222.3 | 206.5 | | 0.0 | 167.2 | 373.7 |
| **Total** | **4.1** | **7.7** | **-12.4** | **-378.7** | **405.0** | **-75.4** | **50.5** | **275.5** | **680.5** |

| Date | Modelled latent heat (KJ m$^{-2}$) | Modelled sensible heat (KJ m$^{-2}$) | Modelled heat conductive flux (KJ m$^{-2}$) | Observe net longwave radiation (KJ m$^{-2}$) | Absorbed solar radiation without rain (KJ m$^{-2}$) | Energy required to reach the isothermal state (KJ m$^{-2}$) | Heat brought into the snow by rain(KJ m$^{-2}$) (Direct heat input + latent heat) | Absorbed solar radiation due to reduced albedo by rain(KJ m$^{-2}$) | Observed net solar radiation(KJ m$^{-2}$) |
|---|---|---|---|---|---|---|---|---|---|
| 5/25/2017 4:00 | -11.5 | -9.6 | -2.8 | -22.0 | 7.5 | | 33.0 | 6.2 | 13.7 |
| 5/25/2017 5:00 | -5.8 | -3.3 | -2.4 | -15.9 | 14.9 | | 33.0 | 9.5 | 24.3 |
| 5/25/2017 6:00 | 0.1 | 1.1 | -2.6 | -1.0 | 29.9 | | 0.0 | 16.3 | 46.2 |
| 5/25/2017 7:00 | -8.7 | -11.3 | -2.9 | -21.9 | 25.4 | | 131.8 | 21.4 | 46.9 |
| 5/25/2017 8:00 | -10.4 | -14.0 | -3.4 | -22.4 | 50.3 | | 0.0 | 43.8 | 94.1 |
| **Total** | **-36.3** | **-37.1** | **-14.0** | **-83.2** | **128.0** | **-109.7** | **197.7** | **97.1** | **225.1** |





| Date | Modelled latent heat (KJ m$^{-2}$) | Modelled sensible heat (KJ m$^{-2}$) | Modelled heat conductive flux (KJ m$^{-2}$) | Observe net longwave radiation (KJ m$^{-2}$) | Absorbed solar radiation without rain (KJ m$^{-2}$) | Energy required to reach the isothermal state (KJ m$^{-2}$) | Heat brought into the snow by rain(KJ m$^{-2}$) (Direct heat input + latent heat) | Absorbed solar radiation due to reduced albedo by rain(KJ m$^{-2}$) | Observed net solar radiation(KJ m$^{-2}$) |
|---|---|---|---|---|---|---|---|---|---|
| 5/27/2017 5:00 | 8.6 | 22.2 | 5.2 | -136.9 | 38.1 | | 29.3 | 26.4 | 64.5 |
| 5/27/2017 6:00 | 1.2 | 2.9 | 5.8 | -163.9 | 33.7 | | 150.5 | 33.5 | 67.2 |
| 5/27/2017 7:00 | 6.7 | 15.4 | 5.8 | -153.5 | 99.5 | | 0.0 | 55.8 | 155.3 |
| 5/27/2017 8:00 | 7.1 | 26.0 | 2.4 | -151.7 | 156.3 | | 0.0 | 118.4 | 274.7 |
| 5/27/2017 9:00 | 7.5 | 24.1 | 1.4 | -188.6 | 150.9 | | 138.7 | 156.2 | 307.1 |
| **Total** | **31.1** | **90.6** | **20.5** | **-794.5** | **478.5** | **-449.5** | **318.5** | **390.4** | **868.9** |





**Figure 1: Rainfall and variations in air temperature and snow depth recorded near Pt. Barrow, Alaska. The data was observed at MB site on Chukchi Sea landfast ice between January and June in 2013, 2014 and 2015. Amount (mm-we) and timing of rainfall are indicated by blue triangles.**





**Figure 2: Observed air temperature, snow depth and liquid precipitation over coastal Chukchi Sea ice in 2000, 2001, 2002, 2006, 2009, 2016.**





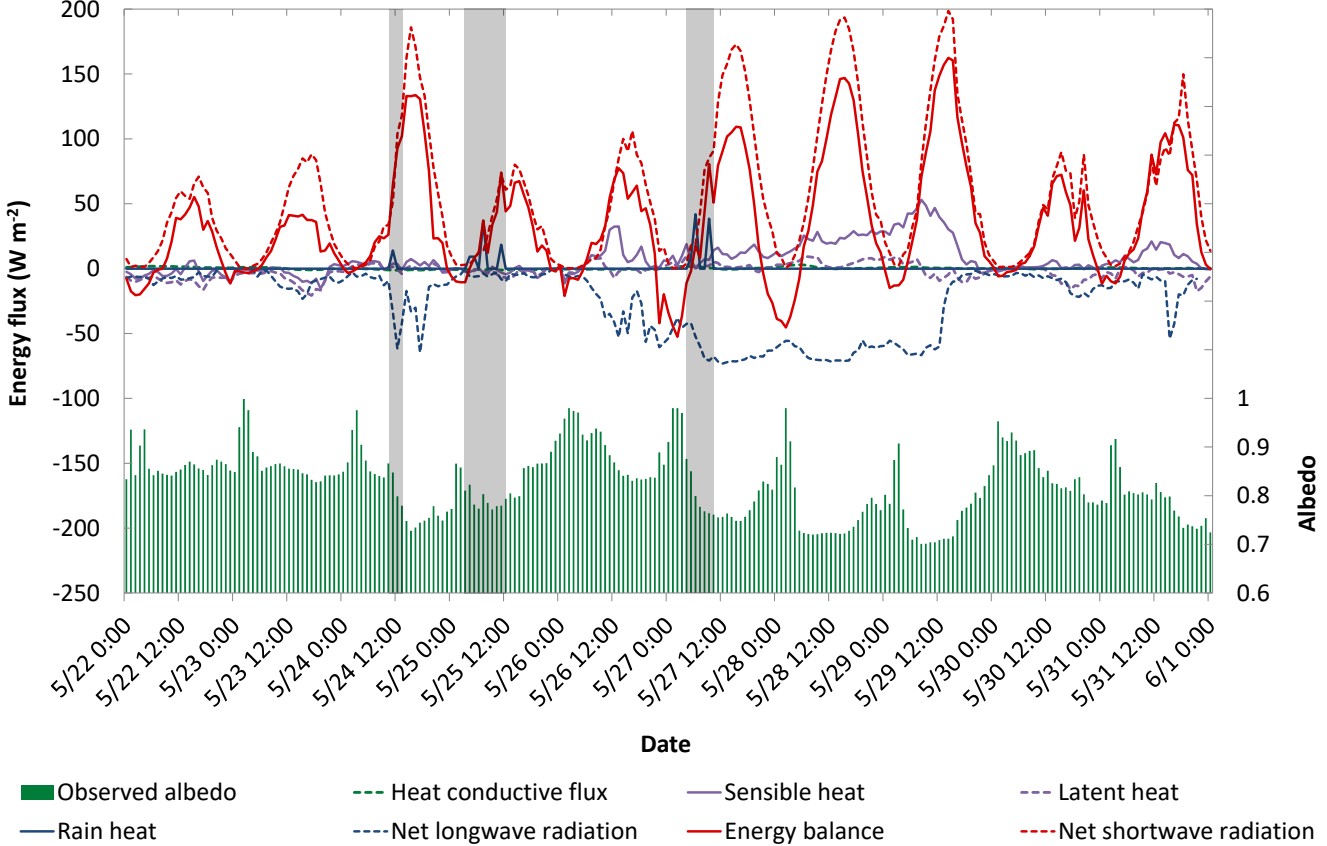

**Figure 3: Energy balance of snow over sea ice during the early stage of melt season. Observed net solar radiation, albedo, net longwave radiation and timing of rainfall from May 22nd through June 1st, 2017 (local time). Calculated sensible heat, latent heat, heat conductive flux and energy budget during the same period are also shown. Rain heat includes the heat that rain directly brings into the snowpack and the latent heat release when the rain freezes within the snowpack. Gray shading shows the timing of rainfall.**



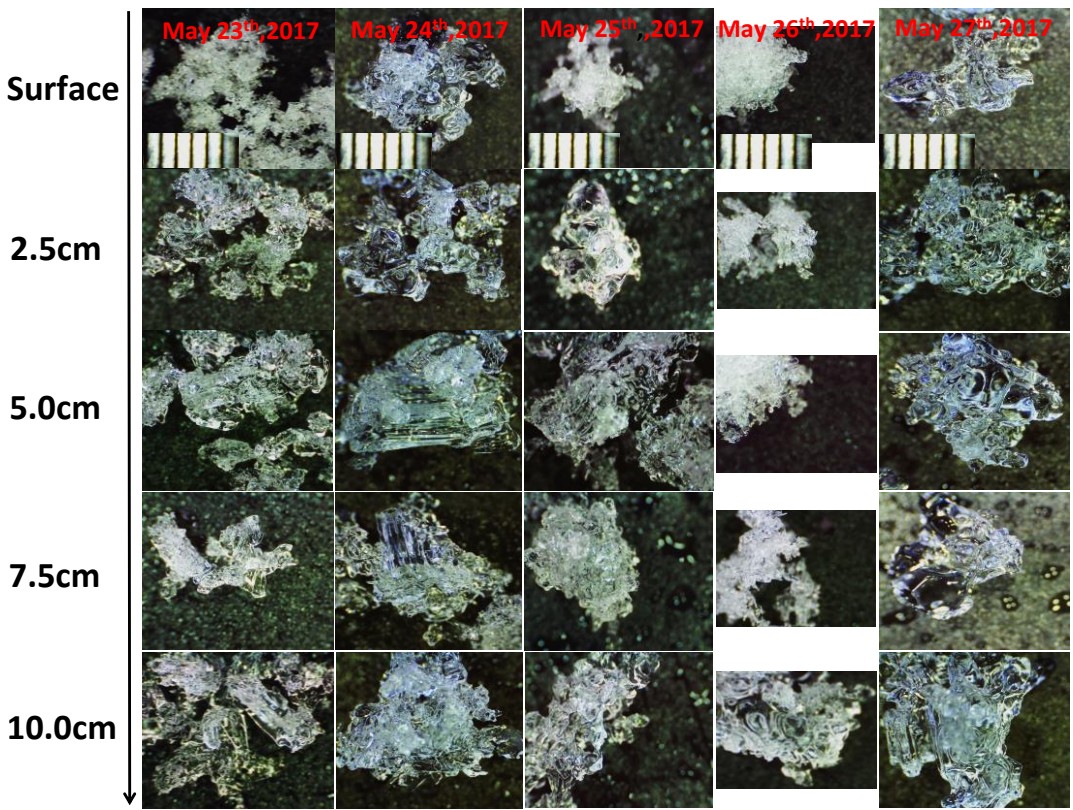

**Figure 4: Observed snow morphology at different depths of the snowpack over Chukchi sea ice north of Utqiaġvik from May 23rd through May 27th, 2017. The reference ruler is 0.5mm long.**





**Figure 5: Observed and modeled snow density, thickness and snow water equivalent (SWE) at different stages of snow melt. A sensitivity experiment without rain was conducted for the same time period and the corresponding SWE is also shown. Rainfall, snow density and depth were observed at the surface of Chukchi Sea landfast ice from May 22nd through June 1st, 2017. A detailed description of the ablation process is provided in the "Observations and Model Simulations of Key Processes" section.**





**Figure 6: Comparison of air temperature observed at WSO AP (Utqiaġvik Weather Service Office airport weather station) and MB site during January-June, 2013 (upper panel), 2014 (middle panel), 2015 (bottom panel).**





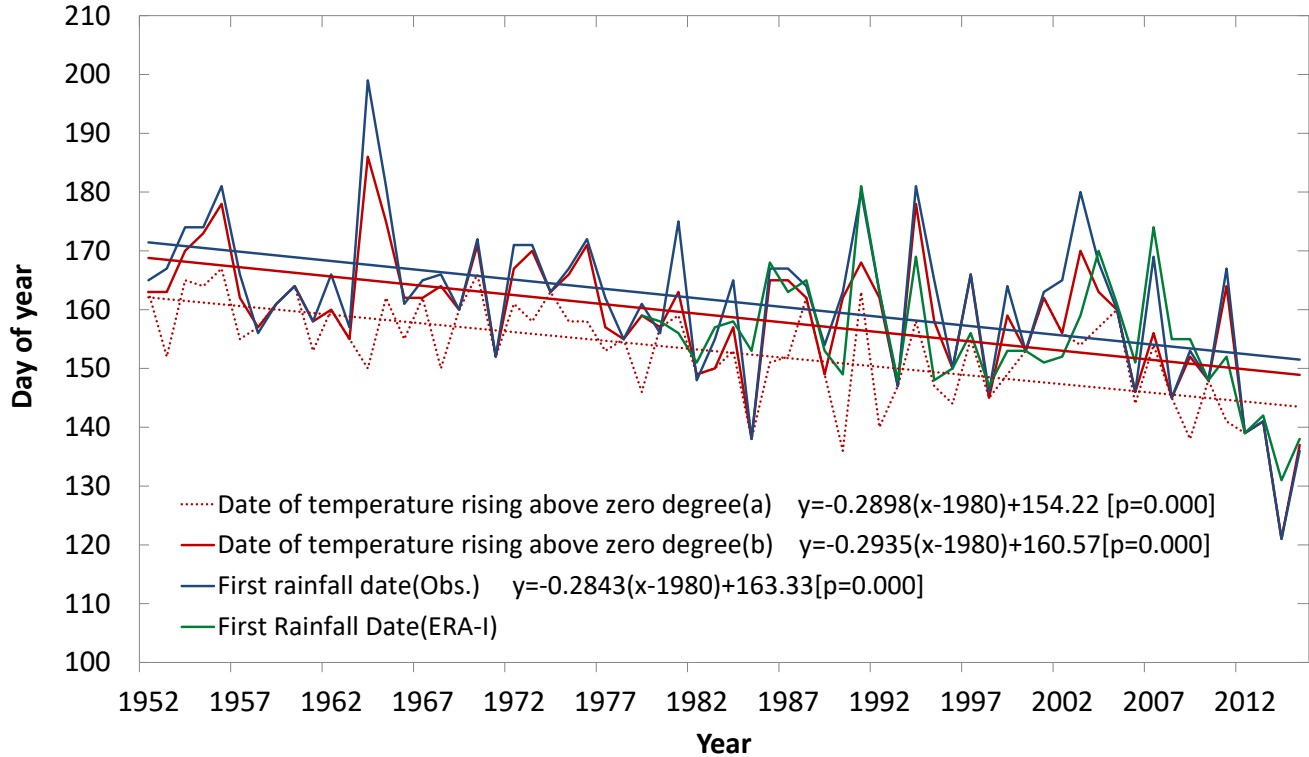

**Figure 7: Timing of air temperature exceeding 0°C for the first time and the timing of first rainfall in spring at WSO AP site, 1952-2015. The red dashed line (a) corresponds to the first instance of air temperature above 0°C. The red solid line (b) indicates the first warming event continuing for at least 4 days. P denotes significance value of the linear trend.**





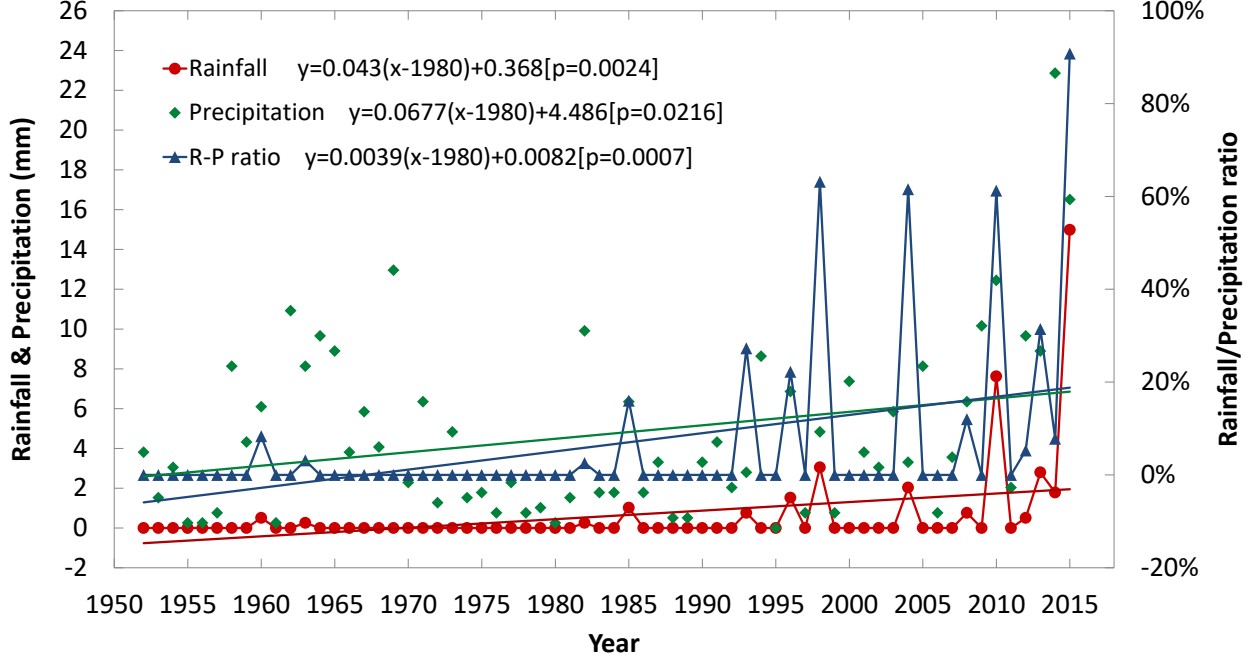

**Figure 8: The variation trend of rainfall, total precipitation, and R-P ratio at Utqiaġvik for May, 1952-2015. P notes significance value of the linear trend.**



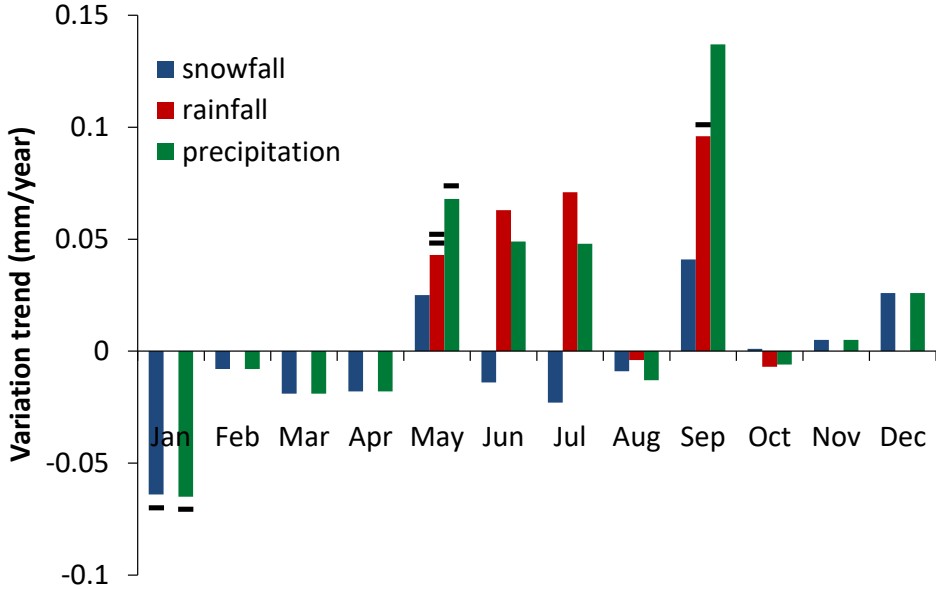

**Figure 9: The variation trends of rainfall, snowfall and precipitation for each month at Utqiaġvik from January 1952 to December 2015. The trend is characterized by the slope of the linear regression equation of the time series.**

5    -    **indicates 0.05 significance level or better**

     = **indicates 0.02 significance level or better**