# Peer review of "A key factor initiating surface ablation of Arctic sea ice: Earlier and increasing liquid precipitation"

_The Cryosphere, 2018_

## Referee Comment (RC1) · Anonymous Referee #1 · 13 Dec 2018

While the topic of the manuscript falls with the preview of the Cryosphere, the ideas are not novel; however, the approach to quantifying, or alt the least qualifying, the impacts of increasing rain on snow events in the late fall and early winter seasons, as well as earlier rain on snow events in the late winter and early spring season in important. This manuscript serves as a simplified starting point for further exploration. However, there are several large gaps in that have escaped consideration, or even mention, in this work. The greatest being the presence of brine in snow over first-year sea ice. Per the extensive literature, brine is known to impact the thermodynamics of snow covers as it allows for melt and the presence of liquid water at temperatures below 0 C, and through the constant phase and volume change in brine pockets. This is not addressed or even

noted as a consideration in this manuscript, and warrants major revisions to define and discuss, at the very least, the theoretical impacts of brine (assuming it wasn't actually measured as a variable in this experiment) and brine volume throughout each step of the analysis, and any and all potential impacts to the conclusions derived from this work.

Additionally, there is no representation at all showing the composition, stratigraphy (layering, intra-pack ice layers), or distribution of either the character (density, etc), or the depth of the snow on first-year sea ice...all of which potentially impact the effect of rain on snow thermodynamics through runoff, percolation, and drainage. There is not even a representation of the character of an "average" snow pack during the case study where only photos of grains are presented. Major revisions are required to address these impacts, or at the very least, acknowledge the theoretical impacts and how they may affect the results and conclusions drawn by this work.

Minor revisions include: 1. Line 390-391: Plot/represent synoptic events in Figure 1 and 2 to better visualize with the rainfall/snow depth/air temp trends. 2. Line 399: Units for water content (ie. %) should be indicated in the text and in Table 1. 3. Line 418: Show a full temperature profile/gradient of the snowpack over time with "average" temperature of -0.7C. 4. Line 434: Show full temperature profile/gradient of the snowpack over time. 5. Introduction: "Here we investigate. . ." There should be specific quantitative questions driving this investigation. What are they? Answer these quantitative questions in the Discussion/Conclusion section of the manuscript? Asking very specific (quantitative/qualitative) questions in the introduction should lead to clearer quantitative/qualitative answers and inferences in the discussion section of this work.

---

## Referee Comment (RC2) · Anonymous Referee #2 · 12 Jan 2019

Title: A key factor initiating surface ablation of Arctic sea ice: Earlier and increasing liquid precipitation Author(s): Tingfeng Dou et al. MS No.: tc-2018-239

General Comments The authors use a model and time series observations from undeformed landfast first-year sea ice to investigate the impact of rain on snow events on sea ice ablation. The authors also use historical rainfall data from a coastal station adjacent to the sea ice cover and find that spring rainfall is occurring earlier, especially since the mid-1990s. The paper addresses a relevant and current topic, seasonal sea ice ablation as it pertains to increased or earlier rainfall contributing to rapid snow ablation due to ripening and decreased albedo.

[Figure]

The authors do a commendable job of incorporating measurements and modelling to explain the impact of rain on snow metamorphism and ablation, as evidence by the agreement between simulations and observations. However as it is presented the results aren't particularly novel, and aside from the rainfall climatology for region of interest, the paper's conclusions about rain on snow are mainly a re-affirmation of the introductory statements (i.e. that these events should likely impact melt pond formation and sea ice ablation). As it is, the snow cover effects are addressed, but impacts on melt pond formation and sea ice ablation are not. Given the location of the study site, the authors should be able to incorporate data on melt pond and sea ice evolution (e.g. pond formation, sea ice thickness, timing of sea ice break-up, etc.) in order to provide valuable insights.

Specific Comments Page = P, Line = L

The title of the paper is perhaps too broad given that the focus is on rain on snow events occurring on an undeformed landfast first year sea ice site.

P2, L7: "... in recent decades."

P3, L1: delete "over sea ice"

P3, L9: Rather than headings for air termperature, wind etc. the appropriate variables should be described under the heading "Micrometeorological Observations". In this section describe air temperature and humidity together with the instrumentation since they were measured and logged together.

P3, L10: state the years of the study in the introductory sentence about the MB site. Section 2.2: Since the model is being presented in detail, include all of the appropriate units (only some are given).

P5, L109: can be shortened to "... snow water equivalent in m, ..."

P5, L113: There is a change to present tense here; be consistent.

P5, L115-118: The temperature profile data should be described in the data section.

Section 3-5 should be "Results" with numbered sub-headings as appropriate.

P6, L29: ". . . rain event of the year . . ."

P7, L1: The rainfall amounts should be also described in the text.

P7, L7: ". . . in surface albedo will result in . . ."

P7, L11-15: Here there are methods that should be described in the methods section.

P8, L8-10: Sentence "A constant SWE and reduced . . . " should be re-written for clarity.

P9, L1-6: Here there are methods that should be described in the methods section.

P10, L16-23: Clarify that this study was for undeformed landfast first year sea ice.

Table 2 is not formatted properly. Heading text could be shortened and footnotes and captions used to explain energy balance components.

---

## Author Comment (AC1) · 6 Feb 2019

We thank the reviewer for a comprehensive and helpful review. The reviewer's comments have guided further improvement in the problem statement and data interpretation. We have also reviewed the relevant literature to further support our central hypothesis and expanded the discussion of study's results. A detailed response follows below.

Major Comments 1) While the topic of the manuscript falls with the preview of the Cryosphere, the ideas are not novel; however, the approach to quantifying, or at the least qualifying, the impacts of increasing rain on snow events in the late fall and early

[Figure]

winter seasons, as well as earlier rain on snow events in the late winter and early spring season in important. This manuscript serves as a simplified starting point for further exploration. However, there are several large gaps in that have escaped consideration, or even mention, in this work. The greatest being the presence of brine in snow over first-year sea ice. Per the extensive literature, brine is known to impact the thermodynamics of snow covers as it allows for melt and the presence of liquid water at temperatures below 0 C, and through the constant phase and volume change in brine pockets. This is not addressed or even noted as a consideration in this manuscript, and warrants major revisions to define and discuss, at the very least, the theoretical impacts of brine (assuming it wasn't actually measured as a variable in this experiment) and brine volume throughout each step of the analysis, and any and all potential impacts to the conclusions derived from this work.

Response: Thank you for your suggestion about the impacts of the physical characteristics of the snow itself on the ablation process, especially mentioning brine-wetted snow which we did not consider before. We have included a discussion about this and depth hoar in the revised MS. From the formation mechanism of the brine, it precipitated from the ice to the snow-ice interface, and then rises in the snow layer by capillary action and reaches a certain height (usually a few centimeters). Brine-wetted snow is found at the base of the snow cover on first-year sea ice. The focus of this paper is how liquid precipitation affects the melting of the snow surface, and the lower part of snow layer are not the focus of this article. However, as the reviewer mentioned, due to high salinity, this phenomenon allows liquid water to exist in the lower part of snowpack at relatively low temperatures. We do not rule out that the presence of brine will accelerate the process of saturation of liquid water in the snow layer, if that is the case. That said, the existence of brine could also accelerate the ripening phase in theory. However, this requires further observations to confirm that. We give a discussion in the revised manuscript, which is also shown as below:

P9-L366-382 (revised MS)ïïjŽ"In addition to the contribution of surface ablation in reducing snow depth, the physical properties of the snow itself will affect the decrease in snow depth to a certain extent when ablation begins. For example, brine may collect at the surface of the sea ice cover as a result of expulsion through surface cracks (Tucker et al., 1992), and will wick into the bottom layers of the snow pack through capillary action. Consequently, the base of the snow pack can consist of such brine-wetted snow (Martin, 1979), with liquid water present even at low temperatures due to the high salinity of the brine (Geldsetzer et al., 2009). Previous work in the Arctic, including at the location studied here, has established that for Arctic snowpack (in contrast with the Antarctic) typically only the lowermost centimetres of the snowpack exhibit higher salt content (Domine et al., 2004; Douglas et al., 2012). Therefore the presence of brine-wetted snow may accelerate the transition of the lowermost snow layers into the ripening phase during ablation, but does not impact the onset of melt in the surface layers of the snowpack.

In addition, in our field work, we found depth hoar to be commonly present at the bottom of the snowpack. Depth hoar is a typical stratigraphic element of the basal layers of the Barrow snowpack during spring season, widely confirmed in previous studies (e.g., Hall et al., 1986; Domine et al., 2012). Depth hoar is conducive to discharge of melt water and subsidence of the snow cover surface, thereby promoting rapid reduction in snow depth. In theory, the presence of both depth hoar and brine-wetted snow supports the rapid reduction of snow depth through the process outlined in this study, though further observations are required to establish the relative importance of this process."

References: Domine, F., Gallet, J. C., Bock, J. and Morin, S.: Structure, specific surface area and thermal conductivity of the snowpack around Barrow, Alaska, J. Geophys. Res., 117, D00R14, 2012. Domine, F., Sparapani, R., Ianniello, A. and Beine, H.J., 2004. The origin of sea salt in snow on Arctic sea ice and in coastal regions. Atmospheric Chemistry and Physics, 4(9/10), pp.2259-2271. Douglas, T. A., et al. (2012), Frost flowers growing in the Arctic ocean-atmosphere–sea ice–snow interface: 1. Chemical composition, J. Geophys. Res., 117, D00R09, doi:10.1029/2011JD016460.

Geldsetzer, T., Langlois, A. and Yackel, J. J.: Dielectric properties of brine-wetted snow on first-year sea ice, Cold Regions Science and Technology, 58, 47-56, 2009. Hall, D. K., Chang, A. T. C. and Foster, J. L.: Detection of the depth-hoar layer in the snow-pack of the Arctic coastal plain of Alaska, U.S.A, using satellite data, Journal of Glaciology, 32, 110, 87-94, 1986. Martin, S.: A field study of brine drainage and oil entrapment in first-year sea ice, Journal of Glaciology, 22, 88, 473–502, 1979. Tucker III, W. B., Perovich, D. K., Gow, A. J., Weeks, W. F. and Drinkwater, M.R.: Physical properties of sea ice relevant to remote sensing. In: Carsey, F. (Ed.), Microwave Remote Sensing of Sea Ice. Geophysical Monograph. American Geophysical Union, pp. 9–28. Chapter 2, 1992.

Major Comments 2) Additionally, there is no representation at all showing the composition, stratigraphy (layering, intra-pack ice layers), or distribution of either the character (density, etc), or the depth of the snow on first-year sea ice...all of which potentially impact the effect of rain on snow thermodynamics through runoff, percolation, and drainage. There is not even a representation of the character of an "average" snow pack during the case study where only photos of grains are presented. Major revisions are required to address these impacts, or at the very least, acknowledge the theoretical impacts and how they may affect the results and conclusions drawn by this work. Response: Some basic snow characteristics were actually shown in the manuscript, including snow depth and density for the case study (Fig. 5). We did not record the detailed stratigraphic characteristics, but the snow grain macrophotographs (Fig. 4) reflect essential information in this regard as well.

We observed depth hoar distributed at the bottom layer of the snowpack during the field measurement. Earlier studies also confirm ubiquitous occurrence of depth hoar at the base of the snow cover on first-year ice during comparable study periods (e.g., Crocker, 1992; Sturm et al., 2002; Langlois et al., 2007). In depth hoar, the shape of the pore space may increase the contact angle and reduce capillary rise of the liquid phase (Jordan et al., 1999). Basal depth hoar is usually associated with a relatively thin brinewetted snow layer (Crocker, 1992), because the height of capillary rise may be limited by low volumes of brine available for wicking, especially at low temperatures. Therefore, in theory, depth hoar can accelerate the decrease of the snow depth during the ablation process and the outputting of the melt water. We have included a discussion of this part in the revised draft, which can also be seen in the response to your first general comment.

Reference: Crocker, G., 1992. Observations of the snow cover on sea ice in the Gulf of Bothnia. International Journal of Remote Sensing, 3 (13), 2433–2445. Sturm, M., Perovich, D.K., Holmgren, J., 2002. Thermal conductivity and heat transfer through the snow on the ice of the Beaufort Sea. Journal of Geophysical Research, 107 (C10) SHE 19-1. Langlois, A., Mundy, C.J., Barber, D.G., 2007. On the winter evolution of snow thermophysical properties over land-fast first-year sea ice. Hydrological Processes, 21 (6), 705–716. Jordan, R. E., Hardy, J. P., Perron, F. E., Fisk, D. J., 1999. Air permeability and capillary rise as measures of the pore structure of snow: an experimental and theoretical study. Hydrological Processes, 13, 1733–1753.

Minor CommentsïijŽ 1. Line 390-391: Plot/represent synoptic events in Figure 1 and 2 to better visualize with the rainfall/snow depth/air temp trends. Response: In this figure, in order to emphasize the corresponding changes in snow depth when the rainfall occurs, we only show the precipitation or weather events associated with the snow melting process. The trends of precipitation and temperature have specially been shown in Fig.7 and Fig.8.

2. Line 399:Units for water content (ie. %) should be indicated in the text and in Table 1. Response: The units of water content of snow is "cm3 water/100 cm3 snow", it has been included in the MS. Please see details in caption in the revised Table 1.

3. Line 418: Show a full temperature profile/gradient of the snowpack over time with "average" temperature of -0.7C. Response: The profile of snow temperature is shown as below, while in the manuscript, we made a brief statement (P7, L316-318 in the

revised MS): "The first rainfall in 2017 was recorded as starting at 10:00am on May 24th (the average snow cover temperature was -0.7°C). The observed snow temperature showed that the upper layers of the snow pack (16cm) became isothermal after two hours." This is clear enough to describe the state of snow temperature from "cold state" to the "isothermal state".

Snow-ice interface -16cm -8cm -2cm Snow surface 5/24/2017 10:00 -1.650 -0.740 -0.430 -0.410 -0.394 5/24/2017 11:00 -1.590 -0.620 -0.410 -0.280 -0.096 5/24/2017 12:00 -1.300 -0.002 0.009 0.010 0.050

4. Line 434: Show full temperature profile/gradient of the snowpack over time. Response: The profile of snow temperature is shown as below, while in the manuscript, we made a brief statement (P7, L329-330 in the revised MS): "From the night of May 24th to the morning of May 25th, the snow temperature fell below the freezing point (-1.0°C), and then rainfall occurred at 4:00 am on May 25th. The snow temperature observations demonstrated that the snow layer reached an isothermal state after five hours."

Snow-ice interface -16cm -8cm -2cm Snow surface 5/25/2017 4:00 -1.230 -0.670 -0.620 -0.530 -0.490 5/25/2017 5:00 -1.230 -0.590 -0.130 -0.490 -0.520 5/25/2017 6:00 -1.260 -0.430 -0.450 -0.450 -0.430 5/25/2017 7:00 -1.230 -0.420 -0.030 -0.120 0.070 5/25/2017 8:00 -1.150 0.001 0.090 0.090 0.150

5. Introduction: "Here we investigate: : :" There should be specific quantitative questions driving this investigation. What are they? Answer these quantitative questions in the Discussion/Conclusion section of the manuscript? Asking very specific (quantitative/qualitative) questions in the introduction should lead to clearer quantitative/qualitative answers and inferences in the discussion section of this work. Response: Thank you for your comments. We have improved the logical connections and statements here (P3, L66-67 in the revised MS). "In order to determine how liquid precipitation affects the surface ablation of sea ice and assess its quantitative

contribution to the reduction in snow depth over sea ice, here,. . .."

Please also note the supplement to this comment:
https://www.the-cryosphere-discuss.net/tc-2018-239/tc-2018-239-AC1-
supplement.pdf

**Supplement:**

[revised manuscript text omitted]

---

## Author Comment (AC2) · 6 Feb 2019

We thank the reviewer for a helpful review. The reviewer's comments have guided further improvements in the logic and statement, making this work more rigorous. A detailed response follows below. Major CommentsïijŽ The authors use a model and time series observations from undeformed landfast first-year sea ice to investigate the impact of rain on snow events on sea ice ablation. The authors also use historical rainfall data from a coastal station adjacent to the sea ice cover and find that spring rainfall is occurring earlier, especially since the mid-1990s. The paper addresses a relevant and current topic, seasonal sea ice ablation as it pertains to increased or

earlier rainfall contributing to rapid snow ablation due to ripening and decreased albedo.

The authors do a commendable job of incorporating measurements and modelling to explain the impact of rain on snow metamorphism and ablation, as evidence by the agreement between simulations and observations. However as it is presented the results aren't particularly novel, and aside from the rainfall climatology for region of interest, the paper's conclusions about rain on snow are mainly a re-affirmation of the introductory statements (i.e. that these events should likely impact melt pond formation and sea ice ablation). As it is, the snow cover effects are addressed, but impacts on melt pond formation and sea ice ablation are not. Given the location of the study site, the authors should be able to incorporate data on melt pond and sea ice evolution (e.g. pond formation, sea ice thickness, timing of sea ice break-up, etc.) in order to provide valuable insights.

Response: This study focused on the effect of liquid precipitation on the early surface melt onset. The subsequent impacts on the melt pond, sea ice evolution will be studied in the next step. We have clarified this at the end of the article (P11, L440 in the revised MS).

Minor CommentsïijŽ The title of the paper is perhaps too broad given that the focus is on rain on snow events occurring on an undeformed landfast first year sea ice site.

Response: We estimated the distribution of liquid precipitation over sea ice during the early melt season using ERA-Interim reanalysis data (Fig. r1), and combined with our previous findings (Han et al., 2018), it can be seen that liquid precipitation occurs over a large area of the Arctic Ocean and is not limited to landfast first year sea ice. According to the conclusion of this paper, liquid precipitation plays a key role in promoting the surface ablation of sea ice, mainly by reducing albedo and releasing latent heat. That is, as long as there is snow on the sea ice surface in the early melt season, the influencing mechanism of liquid precipitation will work, no matter what kind of sea ice is involved. In view of this, we have retained the original title.

Reference: Han W, Xiao C D, Dou T F, et al. Arctic has been going through a transition from solid precipitation to liquid precipitation in spring (in Chinese). Chin. Sci. Bull., 2018, 63, doi: 10.1360/N972018-00088.

P2, L7: ": : : in recent decades." Done.

P3, L1: delete "over sea ice" Done.

P3, L9: Rather than headings for air termperature, wind etc. the appropriate variables should be described under the heading "Micrometeorological Observations". In this section describe air temperature and humidity together with the instrumentation since they were measured and logged together.

Response: Thank you for your comment, we classify observations of meteorological variables, such as air temperature and humidity into one category and use the title of "Micrometeorological Observations at MB Site". The description of snow depth observations is moved to "Radiation, Albedo, Surface Temperature and Snow Depth near MB Site". Details can be seen in "Data section" in the revised MS.

P3, L10: state the years of the study in the introductory sentence about the MB site.

Response: The study period has been included.

Section 2.2: Since the model is being presented in detail, include all of the appropriate units (only some are given).

Response: The units of all variables used in the model have been included. Please see details in the "Modelling of Snow Depth, Snow Density and SWE" in the section of "Methodology".

P5, L109: can be shortened to ": : : snow water equivalent in m, : : :" Done.

P5, L113: There is a change to present tense here; be consistent.

Response: The tense has been revised to be consistent with the previous section.

Please see details in P5, L235-249.

Please also note the supplement to this comment:
https://www.the-cryosphere-discuss.net/tc-2018-239/tc-2018-239-AC2-
supplement.pdf
* * *
b

Fig. r1. The distribution of liquid precipitation (units: m) over Arctic sea ice on May during 1979-2015.

**Fig. 1.**

**Supplement:**

[revised manuscript text omitted]